# Experiences of People Living with Parkinson's Disease in Care Homes: A Qualitative Systematic Review

Shannon Copeland [1], Tara Anderson [1], Gillian Carter [1], Christine Brown Wilson [1], Patrick Stark [1], Mihalis Doumas [2], Matthew Rodger [2], Emma O'Shea [3], Laura Creighton [1], Stephanie Craig [1], James McMahon [1], Arnelle Gillis [1], Sophie Crooks [1] and Gary Mitchell [1,*]

1   School of Nursing and Midwifery, Queen's University Belfast, Belfast BT9 7BL, UK;
    scopeland07@qub.ac.uk (S.C.); tanderson@qub.ac.uk (T.A.); g.carter@qub.ac.uk (G.C.);
    c.brownwilson@qub.ac.uk (C.B.W.); p.stark@qub.ac.uk (P.S.); laura.creighton@qub.ac.uk (L.C.);
    scraig22@qub.ac.uk (S.C.); j.mcmahon@qub.ac.uk (J.M.); agillis03@qub.ac.uk (A.G.);
    scrooks08@qub.ac.uk (S.C.)
2   School of Psychology, Queen's University Belfast, Belfast BT9 7BL, UK; m.doumas@qub.ac.uk (M.D.);
    m.rodger@qub.ac.uk (M.R.)
3   Centre for Gerontology and Rehabilitation, School of Medicine, University College Cork,
    T12 YN60 Cork, Ireland; emma.oshea@ucc.ie
*   Correspondence: gary.mitchell@qub.ac.uk

**Abstract:** Background: Incidence of disability secondary to Parkinson's disease is increasing faster globally than any other neurological condition. The diverse appearance of symptomatology associated with Parkinson's, and the degenerative nature and subsequent functional decline, often increase dependence on caregivers for assistance with daily living, most commonly within a care home setting. Yet, primary literature and evidence synthesis surrounding these unique and complex care needs, challenges and the lived experiences of this population living in long-term nursing or residential facilities remains sparce. The aim of this review is to synthesize qualitative literature about the lived experience of people with Parkinson's disease living in care home settings. Methods: A systematic search of the literature was conducted in October 2023 across six different databases (CINAHL, Medline, EMBASE, PsycINFO, Scopus and Cochrane Library). The Preferred Reporting Items for Systematic Reviews and Meta-Analysis (PRISMA) was used to guide this review. Results: Five articles met the inclusion criteria. Four themes were identified following evidence synthesis: (1) Unique pharmacological challenges. (2) Transitioning and adapting to care home life and routines. (3) Dignified care within care homes. (4) Multidisciplinary care vacuum in care homes. Conclusion: This review revealed the significant and unique challenges for people with Parkinson's disease when transitioning into care homes. These are exacerbated by wider social care challenges such as staffing levels, skill mixes and attitudes as well as a lack of disease-specific knowledge surrounding symptomatology and pharmacology. The lack of multi-disciplinary working and risk-adverse practice inhibited person-centred care and autonomy and reduced the quality of life of people living with Parkinson's disease in care homes. Recommendations for practice highlight training gaps, the need for consistent and improved interdisciplinary working and better person-centred assessment and care delivery.

**Keywords:** Parkinson's disease; neurological disease; care homes; older people; quality of life; experience; health and wellbeing; systematic review

## 1. Introduction

Parkinson's disease (PD) is a complex, chronic, neurodegenerative and multisystem disorder which encompasses a range of diverse and fluctuating motor and non-motor syndromes [1]. Of neurological conditions, which are now the leading cause of disability worldwide, PD is the most rapidly increasing [2]. This exponential rise is principally driven

by an aging population and increased survivorship [3]. There are around 137,000 people living with PD in the United Kingdom (UK) [4]. Furthermore, the Global Burden of Disease Study estimates that the number of cases of PD worldwide will double from around seven million in 2015 to approximately thirteen million in 2040 [5].

PD occurs following dopaminergic cell death in the substantia nigra [6]. Dopamine is a critical catecholamine neurotransmitter in the middle brain which is vital for movement regulation and co-ordination [7]. Therefore, dopamine deficiency is responsible for the cardinal signs of Parkinson's disease: bradykinesia, tremor, rigidity and/or postural instability [8]. Even with optimal therapy, it is inevitable that the symptoms of Parkinson's disease will progress, gradually resulting in disability and reducing the ability to independently perform the activities of daily living, frequently leading to institutionalization [9].

By the time of diagnosis, individuals with PD are often already at an advanced stage of disease and disability, have reduced quality of life, and require complex management [10]. Largely due to the gradual decline in functional status with disease progression, those over the age of 65 diagnosed with PD are likely to live in long-term care supported by healthcare professionals [11]. While there are no reliable reported figures documenting an exact figure of people living in residential care settings in the UK with PD, an American-based study suggested a figure of 5–10% [12]. This number is set to sharply increase in the coming decades [13]. Despite this increasing prevalence of PD globally, and a substantial proportion living within care homes, existing literature has suggested that the care needs of people with PD and their lived experience within care homes remains poorly understood [14,15].

Adapting to care home life can be very challenging for older adults; the transition often entails a sudden change in identity associated with changes in autonomy, daily routine, social status, and contacts [16]. Many older people experience grief reactions to the loss of their home and part of their identity during this transition [17]. Sensory impairments and cognitive deficits can also lead to social isolation while living in a long-term care setting [18]. Specifically for PD, further challenges for care home residents and staff may arise due to the diverse care needs of those with PD, which is characterized by a spectrum of symptoms as well as unique and complex medication regimens. For example, patients' prescriptions often fall outside of standardized, institutional administration times [19]. Individuals with PD also often require a medication dose every one or two hours, which can increase the risk of potential errors [20]. These challenges may be exacerbated in care homes in which there are staff shortages, and a lack of experience in caring for those with PD [21].

In order to develop effective solutions to address such challenges, it is important to first understand the care and support needs from the perspective of those living with the condition [22]. The rise in numbers of older adults in need of nursing home care highlights the need for more research on the lived experience of older adults with degenerative conditions in long-term facilities worldwide [23]. The present review aims to address this by undertaking a qualitative review of the literature. Synthesizing multiple qualitative research studies is a valuable way to extend knowledge and theory [24] by bringing together rich descriptions from multiple perspectives which may not be represented within a single study alone [25].

Therefore, the aim of this review is to synthesize qualitative literature about the lived experiences of people with PD living in care home settings. The two primary objectives were: (1) To collate findings from previous qualitative research conducted in this area to provide a greater understanding of the lived experiences of the population of interest. (2) To appraise the strengths and limitations of the previous literature in order to identify appropriate recommendations for practice and areas where further research is required.

## 2. Materials and Methods

### 2.1. Study Design

A qualitative review was conducted to identify and synthesize existing literature regarding the lived experience of people living with PD in care homes. This was conducted in line with Seers' recommendations regarding qualitative review methodology, following

an integrated review style of synthesis, aggregating data using themes to provide a rich and in-depth understanding [26]. The decision to focus exclusively on qualitative data was driven by the aim of delving into the subjective aspects of the lived experiences of individuals with Parkinson's disease in care homes. Qualitative research allows for a more in-depth exploration of personal narratives, emotions, and contextual factors that quantitative methods may not capture comprehensively. By prioritizing qualitative synthesis, this study aimed to gain a deeper understanding of the multifaceted challenges and perspectives within this specific population, fostering a richer exploration of their lived experience.

### 2.2. Study Selection

A subject librarian was consulted when developing the initial search strategy. The Population Exposure Outcome (PEO) Framework was used to help formulate a feasible research question and identify key and answerable concepts. A thorough search of the literature was conducted in October 2023, using a combination of MeSH terms and keywords such as 'Parkinson' AND 'Care Home' AND 'Experience', as displayed in Table 1, below. Six electronic databases were searched to ensure that all relevant literature could be identified (CINAHL, Medline, EMBASE, PsycINFO, Scopus and Cochrane Library). Selected articles' references lists were also hand-searched to identify potentially relevant works. Further, all 149 full-text articles underwent a supplementary scrutiny process through Google Scholar, wherein their citations were systematically reviewed by the research team to ascertain the absence of any pertinent literature on the subject matter. These additional processes, however, did not yield further studies.

**Table 1.** PEO framework and search terms.

| PEO Framework | Search Terms |
|---|---|
| P: People with Parkinson's disease | Parkinsons or Parkinson's Disease or Parkinsonian, Parkinsonian or PD or Progressive Supranuclear Palsy or Multiple System Atrophy of Corticobasal Degeneration. |
| E: Care home settings | Care homes or care home or Residential homes or residential home or Nursing homes or nursing home or Long term care facilities or long term care facility |
| O: Experience and quality of life | Experience or Outcomes or Quality of life or Perception or Falls or Uncontrollable movements or Cognition or Sleep or Exercise or Mobility or Medications or End-of-life care or Constipation or Infection or Dehydration or Restraint or Dysphagia or Anxiety or Stress. |

### 2.3. Criteria for Inclusion

This review was guided by the Preferred Reporting Items for Systematic Reviews and Meta-Analyses (PRISMA) Statement to identify, screen and ensure eligibility for the inclusion of primary literature [27], as displayed in Figure 1, below. Articles were only included if the study employed a qualitative or mixed methodology, was peer-reviewed and focused explicitly on the lived experiences of Parkinson's disease in care homes as defined as nursing or residential care facilities. Long-term hospital wards, rehabilitation units and skilled nursing facilitates were not included because these settings have different care structures, goals and patient populations compared to care home settings.

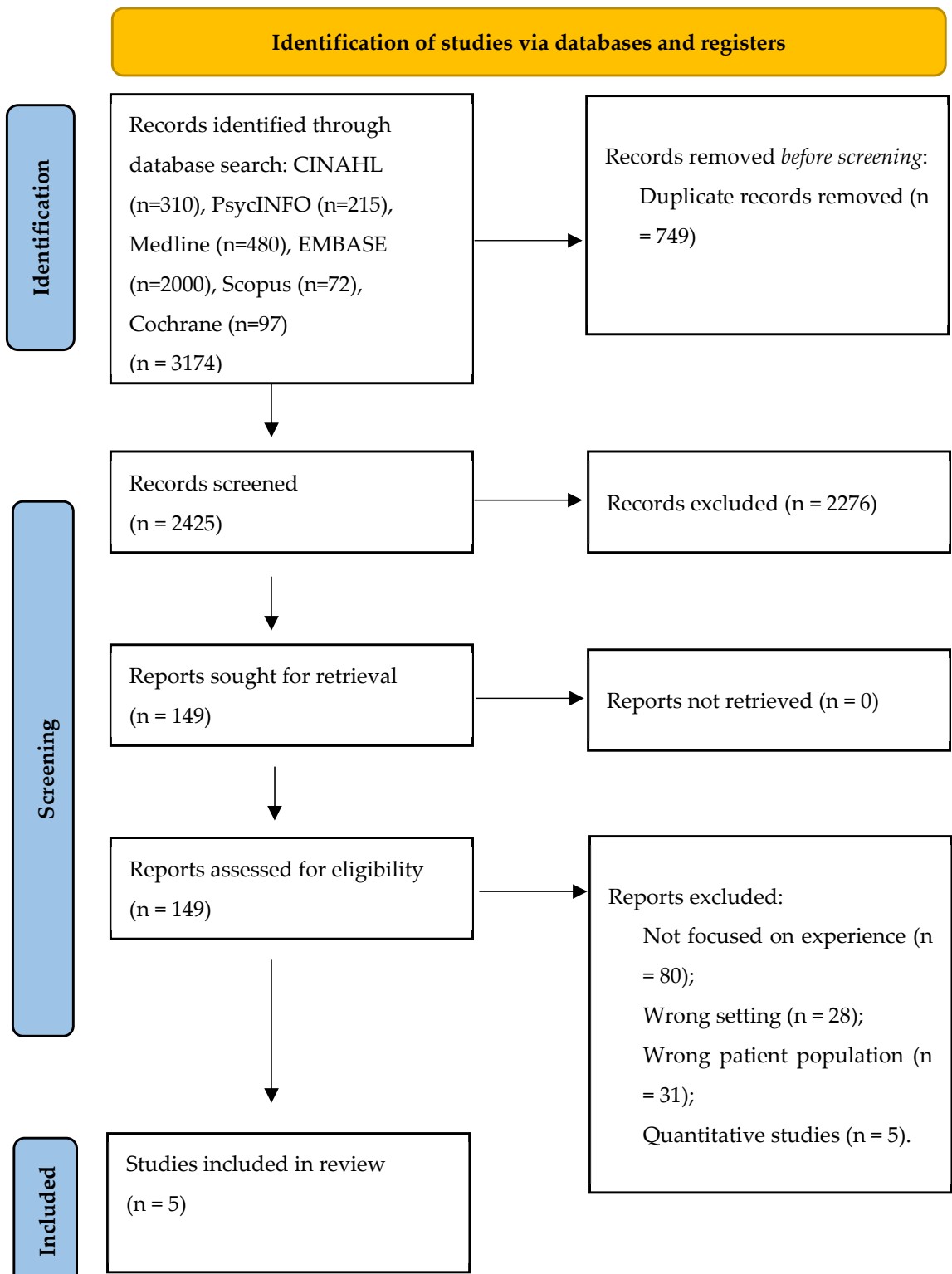

**Figure 1.** Study selection process (PRISMA) flowchart [27].

There were no restrictions for country of publication, but only articles published in English were selected. Experiences could be reported by residents, family members or professional caregivers. Studies were excluded if they were based in a hospital setting or a community dwelling. Thereafter, full texts of potentially relevant studies were reviewed manually to determine eligibility.

The search yielded 3174 results. After removing 749 duplications, the titles and abstracts of 2425 articles were screened and a further 2276 were excluded as they either focused on the wrong experience, the wrong setting or patient population, or used solely quantitative methods. Finally, 149 articles were assessed for eligibility based on the inclusion and exclusion criteria for this review, and 5 studies were selected.

### 2.4. Quality Appraisal

While no studies were to be excluded based on quality, a quality appraisal of all included studies was carried out to aid in the interpretation and synthesis of the findings. The Critical Appraisal Skills Program (CASP)'s checklist for qualitative studies was used to assess the methodological quality and strengths and limitations of primary literature prior to inclusion within this review (https://casp-uk.net/checklists/casp-qualitative-studies-checklist.pdf accessed 18 December 2023). The tool has ten questions, encouraging the researcher to consider whether the research methods selected, such as recruitment, and data collection and analysis, were appropriate and whether subsequent findings, therefore, were credible and meaningful [28,29]. Each paper was scored out of 10, with scores ranging from 7 [30] to 10 [31]. Three studies [32–34] all received a total score of nine, meaning that the studies included within this review were of high methodological quality. A breakdown of the completed checklist for each paper is presented in Table 2.

**Table 2.** Completed CASP checklist for included papers.

| CASP Checklist for Included Papers | | | | | |
|---|---|---|---|---|---|
| | Lex et al., 2018 [30] | Oates et al., 2019 [32] | Armitage et al., 2009 [33] | Fidder et al., 2022 [31] | Van Rummund et al., 2014 [34] |
| Section A | | | | | |
| Q1. Was there a clear statement of the aims of the research? | Yes | Yes | Yes | Yes | Yes |
| Q2. Is a qualitative methodology appropriate? | Yes | Yes | Yes | Yes | Yes |
| Q3. Was the research design appropriate to address the aims of the research? | Yes | Yes | Yes | Yes | Yes |
| Q4. Was the recruitment strategy appropriate to meet the aims of the research? | Yes | Yes | Yes | Yes | Yes |
| Q5. Was the data collected in a way that addressed the research issue? | Cannot tell | Yes | Yes | Yes | Yes |
| Q6. Has the relationship between researcher and participants been adequately considered? | No | No | No | Yes | No |
| Section B | | | | | |
| Q7. Have ethical issues been taken in consideration? | Yes | Yes | Yes | Yes | Yes |
| Q8. Was the data analysis sufficiently rigorous? | Cannot tell | Yes | Yes | Yes | Yes |
| Q9. Is there a clear statement of findings? | Yes | Yes | Yes | Yes | Yes |
| Section C | | | | | |
| Q10. How valuable is the research? | Very | Very | Very | Very | Very |
| Total Score | 7 | 9 | 9 | 10 | 9 |

NB: This table is an adaptation of the Critical Appraisals Skills Programme (CASP) Qualitative Checklist (2018) [28].

### 2.5. Characteristics of Included Studies

The studies included in this review were predominantly based on a qualitative methodology (n = 4) [31–34] with one mixed-methods study [30]. All studies (n = 5) used semi-structured qualitative interviews with people living with PD in care homes, with one study [31] also including informal caregivers, another including close relatives [33] and a third also using focus groups to collect data from informal caregivers, nursing staff and other allied healthcare professionals involved in the care of people living with PD in care homes.

Of the studies that included the following demographic information, (n = 51) participants were living in a nursing home, which may indicate a greater level of need and/or disability than those living within a residential facility (n = 7). Of the articles that provided details of gender, more females (n = 37) than males (n = 21) were studied. Time living within a nursing or long-term care facility ranged from 1 month to 10 years. Disease duration ranged across the studies from 1 to 26 years, while age ranged from 59 to 93 years. All studies (n = 5) were based within Europe: the United Kingdom (n = 2) [32,33], The Netherlands (n = 2) [31,34] and Austria (n = 1) [30]. No studies from low or lower-middle income countries were identified for inclusion. All included studies (n = 5) reported obtaining ethical approval prior to commencing data collection and all studies (n = 5) used non-probability sampling techniques to recruit participants. Table 3 provides an overview of the participant characteristics.

**Table 3.** Participant characteristics.

| Table of Participant Characteristics | | | | | | | | | | |
|---|---|---|---|---|---|---|---|---|---|---|
| | **Gender** | | **Age** | **Hoen and Yahr Stage** | | | **Type of Care Home** | | **Length of Stay at Care Home** | **Months since Diagnosis** |
| | **M** | **F** | | **III** | **V** | **IV** | **NH** | **R** | | |
| Lex et al., (2018) [30] | 4 | 5 | 59–84 | 0 | 3 | 6 | 9 | 0 | 1–7 | 6–20 |
| Oates et al., (2019) [32] | 4 | 6 | 72–93 | 3 | 6 | 1 | 3 | 7 | 1–6 | 4–26 |
| Armitage et al., (2009) [33] | NA | NA | NA | NA | NA | NA | NA | NA | NA | NA |
| Fidder et al., (2022) [31] | 3 | 6 | 63–85 | 3 | 0 | 6 | 9 | 0 | NA | 1–24 |
| Van Rumund et al., (2014) [34] | 10 | 20 | 60–86 | 2 | 15 | 13 | 30 | 0 | 1 month–10 years | 3–26 |

NH = Nursing Home, R = Residential Home, NA = Not Reported. The Hoen and Yahr scale is widely used to describe symptom burden in people with PD. Stage III indicates considerable slowing of body movements with physical dependence, stage IV represents severe disability, but able to walk or stand unaided, and stage V represents wheelchair or bed-bound individuals.

### 2.6. Synthesis

Braun and Clarke's reflexive approach to thematic analysis was used flexibly to facilitate the generation of themes or patterns within the data [35]. All members of the review team met to discuss the emerging themes, and the synthesis of the data remained a characteristically iterative and repetitive process moving between reading primary papers, extracting and synthesizing data in several cycles, and continuously cross-checking themes against the primary papers [36]. This happened alongside ensuring congruence with the interpretation and synthesis of data by retaining the context in which the original data were embedded to avoid misinterpretations of findings from the primary studies. Table 4 provides a summary of included studies.

**Table 4.** Summary of included studies.

| Research Title and Authors | Types of Research | Setting and Country of Research | Aim of Research | Data Collection Methods | Main Findings |
|---|---|---|---|---|---|
| A pilgrim's journey—When Parkinson's disease comes to an end in nursing homes Lex et al., 2018 [30] | Mixed methods; semi-structured interviews | Residential homes, Austria | Gaining empirical data on the nursing demands of residents in late stage of Parkinson's disease being cared for in residential homes | Semi-structured interviews | Future uncertainty or worry, a sense of abandonment by professionals, particularly neurologists, when older adults enter nursing care, and that effective palliative care relies on compassionate nursing and timely medical support. |
| Improving care home life for people with Parkinson's Oates et al., 2019 [32] | Qualitative semi-structured interviews | Care homes, United Kingdom | To explore the decision-making processes at the time of placement and the experiences of care home residents with Parkinson's disease | Semi-structured interviews | Loss of independence, relationships, and functional abilities; challenges in transitioning, adjusting and adapting to life in a care home; and considerations regarding medication timing, control and impact. |
| Caring for persons with Parkinson's disease in care homes: Perceptions of residents and their close relatives, and an associated review of residents' care plans Armitage et al., 2009 [33] | Qualitative interviews | Care homes, United Kingdom | To collect the views of persons with Parkinson's disease and their close relatives in care homes to establish their collective views of the effectiveness of care | Qualitative interviews | Limited comprehension of Parkinson's disease, encompassing medication and functional variations; the impact of care home environment and culture, including the challenges posed by inflexible institutional routines for individuals with Parkinson's disease; and the absence of comprehensive multidisciplinary involvement. |
| Parkinson rehabilitation in nursing homes: A qualitative exploration of the experiences of patients and caregivers Fidder et al., 2022 [31] | Qualitative semi-structured interviews | Nursing homes, The Netherlands | To address the experiences and needs of patients and their caregivers to propose recommendations for improvement | Semi-structured interviews | Autonomy deprivation, encounters with paternalistic practices and reliance on others; challenges arising from inadequate interprofessional communication; and a spectrum of positive and negative encounters in peer interactions. |

**Table 4.** *Cont.*

| Research Title and Authors | Types of Research | Setting and Country of Research | Aim of Research | Data Collection Methods | Main Findings |
|---|---|---|---|---|---|
| Perspectives on Parkinson disease care in Dutch nursing homes Van Rumund et al., 2014 [34] | Qualitative semi-structured interviews and focus groups | Nursing homes, The Netherlands | To analyse the quality of Parkinson's disease care in Dutch nursing homes from the perspectives of residents, caregivers and health care workers | Semi-structured interviews | Inadequate staff empathy and emotional support, insufficient staff expertise in Parkinson's disease matters, including motor fluctuations, medication errors—primarily mistimed levodopa administration—and suboptimal care organization with restricted access to neurologists and Parkinson's disease nurse specialists. |

## 3. Results

Four main themes arose from the studies: (1) Unique pharmacological challenges. (2) Transitioning and adapting to care home life and routines. (3) Dignified care within care homes. (4) Multidisciplinary care vacuums in care homes.

### 3.1. Theme 1: Unique Pharmacological Challenges

Challenges surrounding medication regimens were addressed frequently across all five studies. Residents perceived that good care of their condition was dependent on timely medical administration and therefore subsequent symptom management [30]. Across the studies, some participants felt that their medication was well managed by staff, while many respondents discussed the detrimental effects that delayed or omitted administration had on their functional ability and quality of life, occasionally affecting them for days. During one discussion, the interviewer themselves witnessed a 45 min delay in the provision of medication by care home staff [32].

Even in cases where care home staff possessed knowledge of the importance of timely medication administration in PD, they stated that the high workload limited their ability to adhere to the strict medication schedule associated with best practice [34]. Staff's lack of specialist knowledge surrounding Parkinson's treatment was noted by residents' relatives, who expressed concerns regarding critical medications being '*inappropriately administered*' or '*controlled release preparations being broken to aid swallowing*' [33].

Another relative recalled a time when staff were consistently administering '*normal Sinemet tablets at 7pm and then a controlled release Sinemet an hour later—double the dose—but it was a fight with the staff as they said they must follow the hospital prescription*' [33]. Others found it contradictory that medication had to be administered, and subsequent supervision provided by staff while patients take their tablets due to '*safety regulations*', while they recalled medication being frequently distributed too late or not at all [31].

Although most participants valued the staff's expertise, there were some reports of concern regarding staff competency [31,33]. For example, patients who received duodenal levodopa infusion stated that nursing staff were not always capable of handling the pump [31]. One patient commented: '*Sometimes they ask me what to do [with the pump]. And I'd like to do it myself, but I'm not able to in the morning... In the morning, I can't talk well yet, I can't explain things correctly. And then they ask me: 'Excuse me madam, what are you saying?' That is rather tiring, in the mornings*' [31]. This was a consistent finding within the literature,

with other participants expressing frustration at broader systemic failures, such as the care homes policies [33].

Compared to other care home residents, people living with PD often have complex medication regimens requiring precise timing and dosing to effectively manage symptoms, which may not be as prevalent or as critical in other chronic conditions. Therefore, ensuring accurate medication administration is crucial for maintaining their functional ability and quality of life. The unique and complex medication regimens of individuals with PD present challenges for PD care in care homes which may be heightened by staff shortages, and a lack of experience and specialist knowledge. Across the literature reviewed, it is evident that there is some confusion regarding the appropriate administration of medication between staff and residents, as well as a level of concern from resident's relatives.

*3.2. Theme 2: Transitioning and Adapting to Care Home Life and Routines*

Many participants reported difficulty accepting that the move into care was permanent: *'I kept thinking, when am I going to go home? They kept telling me, you are home, this is your home and every time they said it, I got more and more upset. I didn't want it to be my home'* [34]. Some participants had sold their homes to fund this care home placement and had mixed responses regarding how it felt to hand over control of their finances to a family member [32]. Some portrayed an initial hesitation which eventually became a sense of relief; for example, one woman with PD reported: *'I didn't know if I like this idea, but then I couldn't manage, I could hardly hold money, never mind have it. I just think. . . let her take the burden'* [32]. Contrastingly, a man with PD, in the same study, described his experience as *'hellish, it takes your, a bit of your, manhood away from you, it takes away your independence'* [32].

Interviews also uncovered issues with the physical care home environment, such as small buttons on remote controls or missing support brackets in bathrooms. Many respondents did not feel these were fit for purpose or tailored towards the needs of those with PD and as a result, this led to a reduction in residents' sense of autonomy [31]. Although requiring physical support, many participants were functioning at an intellectual level that demanded more than the standard provision: *'They have what they call activity days; they have quizzes and things like that. But that I find really depressing because although the others, like me, can't be looked after at home, they are all in an advanced stage [of confusion] . . .. I find it very distressing. I said I don't want to be involved. . . I go occasionally just to show willing'* [33]. This may contribute to the challenges participants described surrounding making friends or conversing with other residents who were perhaps living with cognitive impairments, meaning that residents had to rely on limited interactions with busy staff or visitors to meet their social needs [32].

One study explored opinions on the clustering of residents living with PD in specialized units to overcome such challenges [34]. Although some residents experienced heightened anxiety in the presence of those with more severe functional decline, regarding the confrontation of the potential impending severity of disease in their own future [34], clustering would benefit those who, in another study, struggled with residents from different demographics, stating: *'They notice everything, these people. Last night too, someone said: 'madam, you are wriggling so much! Doesn't it tire you?'* When the interviewer asked if it had bothered her to hear this, she responded *'Yes, sometimes. . .'* [31]. However, in the same study, some residents considered encountering other people with PD as a disadvantage, commenting: *'I don't need to be surrounded by Parkinson's all the time'* [31]. In addition, clustering also presents challenges for relatives and care home staff. For example, caregiver participants expressed concerns regarding potential increased travel distance when visiting loved ones [34], while nurses expressed apprehension regarding the physical and mental demands of caring for people with PD, which could lead to burnout [34].

Some relatives felt that the positive aspects of being cared for outweighed the negative aspects of moving into a care home. Positive experiences of care included social aspects such as having company when dining, having a choice of good-quality food and being able to use the garden facilities [31]. For example, a resident's wife reported: *'At home he was on*

*his own and felt lonely: I was still working part-time. I was so worried. Then we moved him into the nursing home. He is so much better here. The nurses look after him well. . . On this ward lives a lady who enjoys playing cards. So, they play cards together. Every day. He enjoys himself'* (Resident's wife) [30]. Another resident's son was interviewed about his father's situation: *'Nurses care for him extremely well, so there is no burden for me that he lives in a nursing home'* [30].

Transitioning to life in a care home can pose challenges for individuals with Parkinson's disease (PD). However, some residents have reported experiencing initial apprehension that later gave way to a sense of relief. Family members, on the other hand, have noted that despite potential drawbacks, the superior quality of care provided in a care home often outweighs the limitations of home care. A deeper understanding of both the benefits and challenges associated with care home living could contribute to enhancing the transition experience for individuals with PD. Therefore, the experiences of individuals with PD in care homes shed light on their unique struggles in accepting permanent placements and adapting to a new environment while managing the progression of their condition.

*3.3. Theme 3: Dignified Care within Care Homes*

Participants across the studies reviewed found that care home routines did not support the timely provision of help for unplanned and often urgent basic activities of daily living. Participants reported feeling *'small'* and *'worthless'* when staff told them that they did not have time or would need to wait [31]. Participants in a second study echoed these findings, adding that staff shortages exacerbated challenges and directly impacted participants' choices of, for example, when they went to bed [33]. Many residents felt part of a routine of being *'dealt with'* when it was convenient for staff [33]. Another resident described *'Having to rely on somebody else for everything, if I need to go to the toilet I have to wait until somebody comes and helps me'* as a *'sort of loss of personal dignity'* which is *'the most difficult thing'* [32].

All five studies agreed that the quality of care was often hampered by high attrition rates of staff, time pressures and a lack of staff. A high reliance on agency staff with little knowledge regarding PD was also cited as adding to these difficulties [32]. There were issues surrounding skill mixing, with highly educated staff being replaced with newly or less qualified staff than the residents had become familiar with [34]. However, many participants were quick to highlight that while they did not blame staff who were under obvious pressure, it did leave residents with feelings of being a burden [32,33]. Participants often couched their comments in something positive, stating that staff are always *'so busy, but do their best'* [32] and *'the staff are marvelous, but they wouldn't wait, couldn't let me get out what I wanted to say'* which led to feelings of being misunderstood, ignored and silenced [33].

One resident discussed bladder problems, which provided an interesting view into staff attitudes, perhaps due to a lack of specific PD knowledge, and a reduced awareness of functional variation: *'They're supposed to get here within 15 min but it can take up to 45 min, I just cannot wait that long. Fine, but when you hear them laughing and then they ask 'well what do you want?' that's the hardest thing to bear'* [33]. This example illustrates a lack of a person-centred care approach through a prioritisation of own social needs over the residents' basic needs.

A lack of awareness of the diversity of PD characteristics may also lead to staff perceiving a resident as *'awkward'* or *'naughty'* [33]. For example, a relative recalled: *'About six weeks ago I went in, and he was dirty, we asked the staff if they could clean him up. . . They said he's being awkward. I learned that he had not made it to the toilet in time. . . Some days he does and others he doesn't'* [33]. Another resident's relative described a scenario in which staff would give her *'dad a cup and forget that he might let go—but not every time, we had a bad episode with one staff member as she literally refused to accept my dad wasn't being naughty'* [33].

There was also considerable evidence that practical tasks were valued over social engagement: *'Carers go passed and say hello, but then they've gone because they have no time to wait for the reply or are embarrassed about the time it takes for him to reply. They might care for his bed, his dressing and bathing, but what about a chat?'* [33]. Other relatives discussed the absence of a 'proactive' culture among care home staff, with one describing this as: *'you*

*have one foot in the grave, the approach here is about box-ticking—he has been fed, washed, and had his coffee'* [33].

It is evident that care homes are substantially impacted by factors such as staff shortages and time pressures and that these can have a detrimental impact on residents. A lack of staff education and training in PD may also account for some of the negative experiences of residents. However, it is important that these factors are addressed, and residents are treated with dignity and respect for both their physical and emotional wellbeing. While dignity in care is a universal concern, the challenge lies in maintaining independence and autonomy for individuals with PD, even as their condition progresses. This aspect is particularly pertinent given the cognitive and physical fluctuations experienced by PD residents, which require tailored care plans to uphold their dignity.

### 3.4. Theme 4: Multidisciplinary Care Vacuum in Care Homes

A lack of multidisciplinary care was evident across the literature, often leading to restrictions placed on residents with PD. Increased functional loss as the disease progressed was linked to a loss of independence; for example, the opportunity to cook or go shopping freely was lost and some felt that care home policy now governed their lives [32]. One resident commented: *'I would like it if. . . I could go out on my own sometimes just for a walk, you've always got to have somebody with you. I have got quite a few friends that I would like just to be able to walk down and visit them instead of them having to come every week to see me, you feel hospitalized'* [32].

To reduce the risk of adverse events, residents were often advised to perform activities under supervision until they could act alone safely. Some participants rejected this advice: *'Yes, I've fallen two times. But then I think, there are so many people who fall every now and then. They are overprotective here, and that's very sweet of course. But, I say, I don't want to spend the rest of my life locked up in a room'* [31]. Risk aversion was also an issue when discussing eating and drinking, with many people with PD also living with secondary dysphagia. A staff lack of specialist knowledge often inhibited people with PD from having foods they enjoyed for fear of choking in the absence of a comprehensive and person-centred assessment and care plan [33].

Some relatives expressed grave concerns regarding people with PD being restricted to a point where they lose the opportunity to mobilize. This was described as hastening their dependence and functional decline due to not only the staffing constraints, as discussed in the previous theme, but also a staff-perceived lack of knowledge and subsequent fear of taking risks [33]. One resident discussed the detrimental consequences this has had: *'I can't walk now; I have a physiotherapist comes every two weeks by private arrangement. Staff are not trained though, so I don't practice the things I do with the physiotherapist'* [33]. Another resident's son expressed that while he understood the degenerative nature of PD, he still had concerns regarding the lack of availability of physiotherapy: *'Yes, it's definitely deteriorated. . . [but] there's a certain inevitability about that, because of age. I'm a little critical of the amount of physiotherapy available... There doesn't seem to be any consistent means of being able to have physiotherapy on a regular basis* [33]. Therefore, there is evidence that when residents and families do manage to access private multidisciplinary input such as physio, they may not be supported in engaging in the prescribed healthcare interventions.

Other participants discussed the benefits of physiotherapy which they had arranged privately: *'when you come into care. . . You're not deemed to be in hospital and not deemed to be in the community so the domiciliary of physiotherapists is not available to you; now I pay for physiotherapy once per week....it has worked wonders'* [33]. However, when it was accessible, rehabilitation in nursing homes was described as a positive experience: *'Instead of shuffling, I am now able to take proper steps. . . You feel as if you go from being a child to becoming yourself again'* [31].

Medical care provided by geriatricians and neurologists was also discussed across several of the included studies. Some family members felt that these medical professionals *'abandoned'* and no longer cared about older people with PD following admission into care

homes [30], while nursing home physician specialists who desired neurological or geriatrician assistance in managing complex PD drug regimens often experienced difficulties getting in touch with PD-specialized doctors [34]. From the literature reviewed, it appears that some residents seek out private physiotherapy and find this beneficial. Residents and care home staff may benefit from multidisciplinary input such as physiotherapy and neurology. This may especially be the case for PD due to the previously discussed spectrum of PD symptoms and medication routines. The lack of multidisciplinary care coordination and access to specialised services therefore poses significant challenges for PD residents in maintaining their functional abilities and quality of life. This aspect highlights the unique needs of people living with PD in care home settings, which necessitate comprehensive and coordinated care approaches to address their complex symptoms and treatment regimens.

## 4. Discussion

The unique and complex pharmacological regimens, and the spectrum of symptomatology which categorizes PD, can impose many challenges for residents and healthcare professionals [19,20]. In a care home environment in which there is a diverse range of residents with complex needs coupled with a lack of PD-specific staff education and staff shortages, these challenges may be exacerbated [21]. As evidenced across the included studies, these factors have a substantial impact on the experience of care home residents with PD.

PD presents unique pharmacological challenges for care home staff which can lead to negative impacts for residents. Some instances of inappropriate medication administration were identified in this review, including delays [32] and the crushing of tablets [33]. These practices may result in incorrect dosing, affecting clinical outcomes, and have been largely attributed to staff shortages and a lack of experience [21]. While there have been a number of initiatives in the UK to address these issues, such as the 'Get It On Time' campaign launched by Parkinson's UK, there are clear indications that further work is still required [37]. The development of clear self-medication policies has been suggested to ensure those patients who are confident and competent to take responsibility for administering their own medications are supported to do so, potentially reducing omissions or delays through system failures [38]. This may help to address some relative's concerns highlighted in this review regarding staff supervision of medication [31,33].

The challenges surrounding transitioning into long-term care and the difficulties of socializing and subsequent loneliness were frequently discussed within the findings. Participants reported changes in autonomy and routine [31,32], as well as social isolation [32,33], in line with previous works in the literature [16–18]. Non-pharmacological interventions have been suggested for mood disorders and are recommended as the first line of treatment in PD [38]. Although clustering residents with PD in specialized units has been suggested to help overcome these challenges, residents, their relatives and staff have raised some concerns regarding this approach because it can be seen as a form of segregation [34]. These concerns also included increased travel distance for relatives and staff burnout [34]. Non-pharmacological interventions have been suggested to help improve the wellbeing of people with PD [39]. These interventions include physical activity programs tailored to the needs of PD patients, such as specific exercises targeting mobility and balance. Additionally, cognitive skills training has demonstrated effectiveness in addressing cognitive impairments often associated with PD, providing residents with strategies to enhance memory and executive function [40]. Psychological therapies such as mindfulness, cognitive behavioural therapy (CBT) and stress management techniques offer avenues for managing emotional and psychological symptoms commonly experienced by individuals with PD [40]. Incorporating these interventions into care home routines can contribute to enhancing the overall quality of life for residents with Parkinson's disease.

Hierarchal decision-making and a risk-adverse culture within care homes may have contributed to poor person-centred care planning and delivery. Medical models of care are not only reductionist and insufficient in meeting the complex needs of people with

PD, but also neglect a person's independence and dignity and impact their sense of personhood [41,42]. This was described by the residents who recalled being infantilized and labelled as 'naughty', or 'awkward', for example [33]. Residents also discussed their lack of autonomy in choosing, for example, when to go to bed, which has an impact on their dignity [31]. Although residents and their relatives understood the pressures staff face, they did discuss the value of a more proactive culture [33]. A holistic care model which recognizes biological, social, psychological and spiritual needs, and promotes respect, equality and mutuality between care recipients and providers, may help to improve these experiences [43].

This review also highlighted a perceived lack of accessible multidisciplinary care, with residents and their relatives reporting a poor availability of physiotherapy, neurology services and speech and language therapists [30,33]. Recent management guidelines such as those from the UK National Institute for Health and Clinical Excellence (NICE) [44] and the European Physiotherapy Guidelines for Parkinson's Disease [45] have supported physiotherapy in PD care. Physiotherapy has been found to help maximize functional ability and reduce secondary complications through movement [46]. Despite this, referral rates have been historically low due to a poor knowledge of improved outcomes and poor availability of physiotherapy services [47]. In terms of neurology care, one study reviewed and discussed how neurologists often 'lose track' of the 20–40% of patients with PD who are admitted to long-term care [34]. This is consistent among the wider literature, where people with Parkinson's often report falling through service gaps as a result of disjointed care planning and under-resourced services [48].

Participants reported a desire for neurologist involvement [30,34] which has been found to be associated with fewer hospitalizations, lower rates of stroke and increased survivorship [48].

*Strengths and Limitations*

To our knowledge, this is the first review to illustrate a paucity of evidence in this field. However, this sparsity of literature brings about the main limitation of this review, that being the small number of primary studies used to inform the synthesis. However, the authors of this review ensured that the search strategy was robust and that all relevant databases and reference lists of eligible studies were exhausted.

All papers were based on European populations and therefore the representativeness of findings is limited. While sample sizes in three of the five studies were smaller than generally recommended, evidence synthesis can increase the reliability of findings through the provision of consenting and unified statements and the employment of critical appraisal tools to assess the primary literature.

Furthermore, all five studies stated reaching data saturation, giving reasonable assurance that further data collection would have yielded similar results to the themes that continuously emerged [49].

Looking ahead, future research in this field should aim to address the gaps identified in the current literature by conducting larger-scale studies with more diverse populations, including those outside of Europe. Additionally, employing mixed-methods approaches could provide a deeper understanding of the experiences and perceptions of Parkinson's disease patients and their families in care home settings. Moreover, longitudinal studies could offer insights into how these experiences evolve over time.

## 5. Conclusions

This qualitative review has joined an incredibly important conversation at a time when the incidence of PD is rapidly increasing alongside the exponential growth of an older population. Combined with one of the most challenging periods in its history, the findings from these studies may be of interest to audiences transitioning into residential or nursing care homes beyond the PD community. Therefore, this work will be of value to increase awareness of the challenges faced by those living with PD in care homes,

including the importance of a timely delivery of critical medications and the need for better interdisciplinary working, as well as the need for Parkinson's-specific training.

This review has identified many challenges experienced as a resident in a care home, both unique to living with PD and more generally in terms of issues that many adults within the social care system face. Many challenges at a macro level were identified, such as poor management and leadership, as well as a lack of funding, training and staffing. Micro-level issues resulted from poor staff attitudes, residents' concerns for the future, lack of stimulation, social isolation and a loss of independence. A lack of multidisciplinary working appeared to be at the root of many of the issues discussed, such as knowledge gaps, which preceded concerns regarding functionality and pharmacological interventions. Finding solutions to these issues will require large scale studies, resources and interventions, as well as a large shift in culture within the social care system to ensure the gold standard frameworks which underpin policies and procedures of care, such as person-centeredness.

**Author Contributions:** Conceptualization, S.C. (Shannon Copeland), G.C., C.B.W., P.S. and G.M.; methodology, S.C. (Shannon Copeland), T.A., G.C., C.B.W., P.S. and G.M.; software, S.C. (Shannon Copeland), T.A., S.C. (Sophie Crooks), S.C. (Stephanie Craig), J.M. and G.M.; validation, P.S. and G.M.; formal analysis, S.C. (Shannon Copeland), G.C. and G.M.; data curation, S.C. (Shannon Copeland), T.A., G.C., C.B.W. and G.M.; writing—original draft preparation, S.C. (Shannon Copeland), T.A. and G.M.; writing—review and editing, S.C. (Shannon Copeland), T.A., G.C., C.B.W., P.S., M.D., M.R., E.O., L.C., S.C. (Stephanie Craig), J.M. and G.M.; project administration, S.C. (Shannon Copeland), T.A., G.C., C.B.W., P.S., M.D., M.R., E.O., L.C., S.C. (Stephanie Craig), J.M., A.G., S.C. (Sophie Crooks) and G.M. All authors have read and agreed to the published version of the manuscript.

**Funding:** This research received no external funding.

**Institutional Review Board Statement:** Not applicable.

**Informed Consent Statement:** Not applicable.

**Data Availability Statement:** Not applicable.

**Public Involvement Statement:** No public involvement in any aspect of this research.

**Guidelines and Standards Statement:** This manuscript was drafted against the Preferred Reporting Items for Systematic Reviews and Meta-Analyses (PRISMA) Statement to identify, screen and ensure eligibility for the inclusion of primary literature.

**Acknowledgments:** The team wish to thank Colleen Tierney, Subject Librarian at the School of Nursing and Midwifery, Queen's University Belfast, for her support in developing the initial search strategy.

**Conflicts of Interest:** The authors declare no conflicts of interest.

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
