# Peer review of "Experiences of People Living with Parkinson’s Disease in Care Homes: A Qualitative Systematic Review"

_nursrep, doi:10.3390/nursrep14010033_

Round 1
Reviewer 1 Report
Comments and Suggestions for Authors
The purpose of the manuscript is to synthesize qualitative literature about the lived experience of people with Parkinson's disease living in care home setting. The authors did a good job in framing the importance of better understanding the needs of this growing population and I believe this study will make a worthwhile contribution. I have just a few small comments/questions.
1. missing parenthesis in section 2.2, line 118 after Cochrane Library.
2. Table 2 - CASP checklist for included papers -- Section C. How valuable is the research? -- Responses were 'Yes'. This seems incongruent to the question itself. Were initial responses categorized into Yes/No categories? If so, how so?
3. Line 340-342: sentence beginning "While relatives felt these..." Awkward sentence. Continues from the previous, but since it is stand-alone could be worded a bit better.
4. Lines 474-477. I was surprised to see the suggestion of Doll therapy as a possible solution to the difficulties of those with Parkinson's disease adjusting to the socialization and loneliness of transitioning into a care setting. In particular, I picked up in the review that many of the frustrations stemmed from loss of independence, lack of autonomy, and lack of attention to the specific needs of those with Parkinson's disease that differed from others who are in care settings for other reasons - commonly cognitive impairment. Even in the next paragraph, the authors note that those with Parkinson's disease felt they were being 'infantilized' by the care they were receiving. Suggesting that those with their cognition in tact would benefit from doll therapy seems disjointed.
Author Response
Dear Reviewer, Thank you for this kind and supportive feedback.
We have made the recommended changes to points 1 and 2.
Point 3, we have rewritten the text as follows:
"Transitioning to life in a care home can pose challenges for individuals with Parkin-son's disease (PD). However, some residents have reported experiencing initial appre-hension that later gave way to a sense of relief. Family members, on the other hand, have noted that despite potential drawbacks, the superior quality of care provided in a care home often outweighs the limitations of home care. A deeper understanding of both the benefits and challenges associated with care home living could contribute to enhancing the transition experience for individuals with PD."
Point 4 - we have removed reference to doll therapy and presented a more detailed summary of non-pharm interventions: "
These interventions include physical activity programs tailored to the needs of PD pa-tients, such as specific exercises targeting mobility and balance. Additionally, cognitive skills training has demonstrated effectiveness in addressing cognitive impairments often associated with PD, providing residents with strategies to enhance memory and executive function [40]. Psychological therapies such as mindfulness, cognitive behavioural ther-apy (CBT), and stress management techniques offer avenues for managing emotional and psychological symptoms commonly experienced by individuals with PD [40]. Incorpo-rating these interventions into care home routines can contribute to enhancing the overall quality of life for residents with Parkinson's disease".
These changes have been highlighted in red font.
Thank you for this feedback.
Reviewer 2 Report
Comments and Suggestions for Authors
There are some minor problems in the demonstration of the results of the included studies, that is, the perceptions of the respondents may not be unique to PD patients and their families, and such findings may not be of much significance for improving the service quality of PD patients in nursing homes. It is suggested that these comments should be deleted or reduced and that other topics related to PD should be elaborated in more detail.
Author Response
Thank you for this supportive feedback. We have made some changes.
In theme 1 we have added "PD patients often have complex medication regimens requiring precise timing and dosing to effectively manage symptoms, which may not be as prevalent or as critical in other chronic conditions. Therefore, ensuring accurate medication administration is crucial for maintaining their functional ability and quality of life."
In theme 2 we have added "Therefore, the experiences of individuals with PD in care homes shed light on their unique struggles in accepting permanent placement and adapting to a new environment while managing the progression of their condition."
In theme 3 we have added "While dignity in care is a universal concern, the challenge lies in maintaining independence and autonomy for individuals with PD, even as their condition progresses. This aspect is particularly pertinent given the cognitive and physical fluctuations experienced by PD residents, which require tailored care plans to uphold their dignity."
In theme 4 we have added "The lack of multidisciplinary care coordination and access to specialised services therefore poses significant challenges for PD residents in maintaining their functional abilities and quality of life. This aspect highlights the unique needs of people living with PD in care home settings, which necessitate comprehensive and coordinated care approaches to address their complex symptoms and treatment regimens."
We believe these additions will assist the reader in understanding the unique findings of this review in the context of what is already known. Thank you.
Reviewer 3 Report
Comments and Suggestions for Authors
Dear authors,
Many authors expect a lot of feedback from any reviewer asked to review a manuscript. I would like to start my feedback with saying that this is a solid piece of work, well-executed, and honestly a joy to read. I have a background in neurocognitive diseases, and am pleased with the contents of this paper. It is fitting for the scope of the journal, and it is of interest to the field. All PRISMA steps can be traced back in the text. I'd like to say: accept as is. When work is good as is, reviewers should not be nit-picking and come up with a list of petty things to improve.
However, I do think that the artwork and the way Tables are presented can be improved in order to improve the legibility. This is entirely up to the discretion of the authors.
And you have included 5 studied, of which 2 are British, 2 are Dutch and 1 is Austrian. How representative are your findings outside of the European continent? This is something you may wish to discuss, and make recommendations about for future research.
Overall, a well-written paper that should be accepted without much further ado.
Author Response
Thank you for this supportive feedback. We have added a short paragraph in that provides recommendations for future research:
"Looking ahead, future research in this field should aim to address the gaps identified in the current literature by conducting larger-scale studies with more diverse populations, including those outside of Europe. Additionally, employing mixed-methods approaches could provide a deeper understanding of the experiences and perceptions of Parkinson's disease patients and their families in care home settings. Moreover, longitudinal studies could offer insights into how these experiences evolve over time."